# Nitric Oxide Extends the Postharvest Life of Water Bamboo Shoots Partly by Maintaining Mitochondrial Structure and Energy Metabolism

**DOI:** 10.3390/ijms23031607

**Published:** 2022-01-30

**Authors:** Chunlu Qian, Zhengjie Ji, Chen Lin, Man Zhang, Jixian Zhang, Juan Kan, Jun Liu, Changhai Jin, Lixia Xiao, Xiaohua Qi

**Affiliations:** 1Department of Food Science and Engineering, School of Food Science and Engineering, Yangzhou University, Yangzhou 225127, China; miranda_jzj@163.com (Z.J.); lcdeyoux@163.com (C.L.); mzhang@yzu.edu.cn (M.Z.); zjx@yzu.edu.cn (J.Z.); kanjuan@yzu.edu.cn (J.K.); junliu@yzu.edu.cn (J.L.); chjin@yzu.edu.cn (C.J.); lxxiao@yzu.edu.cn (L.X.); 2Department of Horticulture, School of Horticulture and Plant Protection, Yangzhou University, Yangzhou 225009, China

**Keywords:** energy metabolism, modified atmosphere packaging, nitric oxide, postharvest storage, water bamboo shoots, *Zizania latifolia*

## Abstract

Harvested water bamboo shoots can be stored for only a few days before they lose weight and become soft. Nitrogen oxide (NO) and modified atmosphere packaging (MAP) have previously been used to prolong horticultural crop storage. In the present study, we analyzed the joint effect of these two methods on extending the postharvest quality of water bamboo shoots. Water bamboo shoots were treated with (1) 30 μL L^−1^ NO, (2) MAP, and (3) a combination of NO and MAP. The NO treatment delayed the softness and weight loss through maintaining the integrity of the mitochondrial ultrastructure and enhancing the ATP level by activating the expressions and activities of succinic dehydrogenase, malic acid dehydrogenase, and cytochrome oxidase. MAP improved the effect of NO on the mitochondrial energy metabolism. These results indicate that NO and MAP treatments are effective at suppressing the quality deterioration of water bamboo shoots, MAP improves the effect of NO in extending postharvest life, and NO may be the main effective factor in the combination of NO and MAP.

## 1. Introduction

Water bamboo (*Zizania latifolia*; family Gramineae) has been cultivated for more than 2000 years in Southeast Asia (water bamboo is different from the more common bamboo (genus species family)) [1]. In China, water bamboo is a popular aquatic vegetable because of its swollen tender shoot, which is rich in dietary fiber and phytonutrients [2]. However, postharvest quality quickly declines during storage, and water bamboo shoots can only be stored for 2–3 d under normal temperature and 7 d under cold storage. The shoots easily become soft, lose water, and change colors during storage [3].

Postharvest senescence and quality deterioration in vegetables are closely associated with reduced adenosine triphosphate (ATP) levels, which may cause a loss of integrity in biological membranes [4,5]. The tricarboxyfic-acid (TCA) cycle and mitochondrial electron transporting through the cytochrome pathway are the main pathways of plant respiration, which generates ATPs. Succinic dehydrogenase (SDH) and malic acid dehydrogenase (MDH) are key enzymes in the TCA cycle. SDH reversibly oxidizes succinic acid to fumaric acid, and MDH catalyzes the reversible conversion between malic acid and oxaloacetic acid [6]. Cytochrome oxidase (CCO), which catalyzes the transfer of electrons from ferrocytochrome c to molecular oxygen, playing an important role in cytochrome pathway-related plant respiration [7].

Nitric oxide (NO) is a highly reactive gas and has been applied to improve the postharvest quality of peach [8], lettuce [9], bamboo shoots [10], apple [11], tomato [12], and winter jujube [13]. NO was also effective for alleviating the lignification and softening of water bamboo shoots [3]. In addition, some methods including high hydrostatic pressure [14], 1-methylcyclopropene [15], and a combination of low temperature and modified atmosphere packaging (MAP) have also been used to extend the shelf life of water bamboo shoots [16]. MAP in combination with low temperature is a widely acceptable and applicable technology for the maintenance of the postharvest quality of fruits and vegetables [17]. MAP slows down the respiration rate, ethylene production, and weight loss of fruits or vegetables by reducing mitochondrial energy metabolism and antioxidant enzyme activity [18]. However, the mechanisms of MAP and NO that improve the postharvest quality of water bamboo shoots are still unclear.

In the present study, we aimed to investigate the mechanisms of NO and MAP on extending the postharvest life of water bamboo shoots. Firstly, the morphological change, respiration rate, firmness, and weight loss rate were compared among the control, MAP treatment, NO treatment, and a combination of MAP and NO treatments. Then, the ATP contents changes and the energy charge levels (a function of ATP, ADP, and AMP levels) were analyzed. Finally, the ultrastructure of the mitochondrial was observed and the expressions and activities of key enzymes in the TCA and cytochrome pathways were analyzed in water bamboo shoots stored at 4 °C.

## 2. Results

### 2.1. Effects of NO and MAP on Appearance, Weight Loss, and Firmness of Water Bamboo Shoots during Cold Storage

After 14 d of cold storage, the control shoots showed some wilting while the three treated groups showed no obvious wilting. Before storage, the shoots were white and bright. During cold storage, the control and MAP-treated shoots became slightly green while the NO and NO + MAP-treated shoots showed no obvious color changes. After 28 days of storage, the NO + MAP-treated shoots showed the least wilting and skin greening (Figure 1).

The weight loss rates increased in all materials and was significantly (*p* < 0.05) higher in the control and NO-treated groups than in the MAP and NO + MAP groups (Figure 2a). NO treatment restrained the increase in weight loss rate in the material in or out of MAP. The firmness of the shoots in all groups decreased with increasing storage time (Figure 2b). The decrease in firmness was greatest in the control group and lowest in the NO + MAP group, and no significant (*p* > 0.05) difference appeared between the NO and MAP shoots.

### 2.2. Effects of NO and MAP on Respiration Rate of Water Bamboo Shoots during Cold Storage

The respiration rate of all water bamboo shoots increased and peaked after 14 d of cold storage, except for the MAP-treated shoots, which peaked after 21 d of cold storage (Figure 2c). The respiration rate was the lowest in the NO + MAP group during cold storage. However, after four weeks, the respiration rate fell to similar levels as the other groups. NO treatment significantly (*p* < 0.05) restrained the increase in respiration rate. MAP could further reduce the respiration rate in the water bamboo shoot after NO treatment during cold storage but a single MAP treatment did not show any inhibition effect on respiration.

### 2.3. Effects of NO and MAP on ATP and Energy Charge Levels of Water Bamboo Shoots during Cold Storage

ATP content (Figure 3a) and energy charge level (Figure 3b) appeared with a similar change pattern and tended to decrease with the storage time extending in all the groups. However, the decrease was greatest in the control group and lowest in the NO + MAP group. The ATP content and energy charge level in the MAP water bamboo shoot declined during the first 7 d of cold storage and then increased and peaked at 14 d; however, in NO-treated material, they maintained high value during the first 7 d of cold storage and then declined to the end of storage. MAP could further maintain a high level of ATP content and energy charge in the NO-treated water bamboo shoot, and the NO + MAP group exhibited a similar change pattern to the single NO group.

### 2.4. Effects of NO and MAP on Mitochondrial Ultrastructure of Water Bamboo Shoots during Cold Storage 

After 28 d of cold storage, the mitochondria of control shoots were greatly swollen, their cristae had almost completely disappeared, and the inner membrane was extensively ruptured (Figure 4A). The mitochondria of the NO-treated shoots swelled, the cristae partially disappeared, and the inner membranes partially ruptured (Figure 4B). In contrast, the mitochondria of the MAP-treated shoots were structurally intact and the cristae were relatively clear, although the intracristae space was enlarged (Figure 4C). The mitochondria of the NO + MAP-treated shoots showed distinct cristae and integrated structure along with a wide intracristae surface area (Figure 4D).

### 2.5. Effects of NO and MAP on ATPase Activity and Ca^2+^ Content of Water Bamboo Shoots during Cold Storage

The activities of H^+^-ATPase and Ca^2+^-ATPase in the four groups decreased during cold storage (Figure 5a,c). The decrease was greatest in the control group and lowest in the NO + MAP group. The MAP treatment maintained a higher level of H^+^-ATPase and Ca^2+^-ATPase activities in the water bamboo shoots than the NO treatment during the early stage of cold storage, while during the late stage of cold storage, the NO treatment exhibited a higher level than MAP. The activities of Na^+^-K^+^-ATPase in the four groups increased to peaks after one week of storage, except for the MAP-treated material, which continuously rose to a peak after two weeks of cold storage (Figure 5b). During the late stage of storage, the Na^+^-K^+^-ATPase activities declined, with the level kept the lowest in the control group and the highest in the NO + MAP group, and no significant (*p* > 0.05) difference was revealed between the MAP and NO groups.

The content of Ca^2+^ decreased in the water bamboo shoots after 7 d of cold storage, and an obvious increase to peaks was exhibited after 14 d of cold storage, with the exception of the NO + MAP group, which showed a continuous increase. The control group possessed the lowest Ca^2+^ content and the highest level was observed in the NO + MAP group (Figure 5d). MAP-treated water bamboo shoots exhibited higher Ca^2+^ content than the NO-treated material during the whole storage period.

### 2.6. Effects of NO and MAP on Enzyme Activities of MDH, SDH, and CCO of Water Bamboo Shoots during Cold Storage

The MDH activity in all materials increased to obvious peaks after 7 d of cold storage, except for the NO-treated shoots, which continuously rose and peaked at the 14th day (Figure 6a). The peak of MDH activity in the NO + MAP group was significantly (*p* < 0.05) higher than the other material and maintained relative higher activity than other groups to the end of storage, and the control group showed relatively lower activity during the late stage of storage. The SDH activity in all groups increased to peaks after 7 d of cold storage and then decreased, except for the control material, which maintained stable activity and lasted for 21 d of storage, then decreased (Figure 5b). During the early stage of storage, the control exhibited significantly (*p* < 0.05) lower SDH activity than others, while the NO + MAP shoots possessed the highest, and no significant (*p* > 0.05) difference was shown between the MAP and NO groups. The activities of CCO decreased in all groups during storage, with the highest level in the NO + MAP group and the lowest in the control, and no significant (*p* > 0.05) difference was shown between the MAP and NO groups (Figure 6c).

### 2.7. Effects of NO and MAP on the Expression of ZlH^+^-ATPase, ZlNa^+^-K^+^-ATPase, ZlCa^2+^-ATPase, ZlMDH, ZlSDH, and ZlCCO of Water Bamboo Shoots during Cold Storage

The expression level of *ZlH^+^-ATPase*, *ZlCa^2+^-ATPase*, *Zl**CCO* decreased in the water bamboo shoots during cold storage (Figure 7a,c,f). The NO + MAP group exhibited the highest expression level of these three genes while the control group showed the lowest expression levels of *ZlCa^2+^-ATPase*, *Zl**CCO genes*. The MAP-treated water bamboo shoots showed the lowest *ZlH^+^-ATPase* expression level, and no significant (*p* > 0.05) difference was exhibited between the MAP- and NO-treated shoots on the expression of *ZlCa^2+^-ATPase*. The NO-treated shoots possessed a significantly (*p* < 0.05) higher expression level of *ZlH^+^-ATPase*, *Zl**CCO* than the MAP-treated shoots during the whole storage period.

The expression level of *ZlNa^+^-K^+^-ATPase, ZlMDH, ZlSDH* increased and peaked after 7 d of cold storage and then decreased (Figure 7b,d,e), except for the MAP group, which peaked in *ZlNa^+^-K^+^-ATPase* expression, and the NO group, which peaked in *ZlMDH* expression after 14 d of storage. The control shoots exhibited the lowest expression level of *ZlNa^+^-K^+^-ATPase*, *ZlMDH, ZlSDH* during the late stage of storage, and the NO + MAP group showed the highest expression during the early stage of storage. The NO-treated water bamboo shoots expressed a higher expression level of *ZlNa^+^-K^+^-ATPase* than the MAP-treated shoots during the early stage of storage and higher *ZlSDH* during the late stage of storage.

## 3. Discussion

NO is effective at improving the postharvest quality of vegetables and fruits [13,14,15,16,18]. MAP is a technology that alters the atmosphere within the package according to the interaction between the product respiration rate and the transfer of gases through the package [19]. These technologies have been successfully applied to whole and fresh-cut products such as artichokes [20], lettuce [21], strawberry [22,23], persimmon [17], cucumber [24], and water bamboo shoots [16]. To our knowledge, this is the first report on water bamboo shoots demonstrating that the combination of NO and MAP can delay softness and weight loss and enhance postharvest quality (Figure 1 and Figure 2). The preservation effect of NO + MAP on water bamboo shoots was significant and practical.

The important parameters in the postharvest life of water bamboo shoots are weight loss and softness, which are related to respiration rate and energy consumption. NO significantly reduced the softening of water bamboo shoots over the whole storage period (Figure 2). Similar results were found in winter jujube [13], wax apple [11], and peach [8]. In contrast, NO decreased the firmness of bamboo (*Phyllostachys violascens*) shoots [10]. We also showed that the combination of NO and MAP improved the shoot weight loss and firmness compared to individual treatments of NO and MAP (Figure 2a,b). The present results suggested that MAP could significantly restrain the increase in weight loss, NO could restrain the respiration rate, and the combination of NO and MAP could further increase the inhibition effects of each other.

The deterioration of the postharvest quality reflects a profound decline in energy levels [25]. Mitochondria are the main organelle of oxidative phosphorylation, which produces more than 95% of the energy in plants [26,27]. Vegetable senescence in postharvest storage is associated with a decrease in the quantity, structural damage, and dysfunction of mitochondria. In particular, mitochondrial function depends on the integrity of the mitochondrial membrane, and the degradation of the membrane is a major cause of senescence in plants [28]. Ultraviolet-C treatment delayed the senescence of peach fruit by maintaining mitochondrial membrane integrity [29]. Benzothiadiazole treatment inhibits disease in stored apples by maintaining the number and structure of mitochondria [30]. In the present study, the integrity of the mitochondrial structure was maintained by the NO and MAP treatments, and the NO + MAP treatment showed the most integral mitochondrial ultrastructure of water bamboo shoot after cold storage (Figure 4), which implied its normal and healthy function.

Mitochondria are the main engine of ATP, which is the primary energy source for life activities. In plants, MDH, SDH, CCO, Ca^2+^-, and H^+^-ATPase are major enzymes of oxidative phosphorylation and ATP synthesis [31]. The energy status of stored vegetables and fruits is associated with senescence, abiotic resistance, and physiological disorders [6,30,32]. An O_2_/CO_2_ controlled atmosphere delays the senescence of broccoli (*Brassica oleracea* L. var. *italica*) by inhibiting the decrease in ATP level, energy charge, and the activities of SDH and CCO [6]. Similarly, a chlorine dioxide treatment delays the senescence of longan fruit by restoring ATP level, energy charge, and SDH and CCO activities [32]. In the present study, the NO + MAP treatment efficiently increased the activities of enzymes involved in the TCA cycle and electron transport, H^+^-ATPase, Ca^2+^-ATPase, and Na^+^-K^+^-ATPase, which lead to higher ATP levels and energy charges in water bamboo shoots during cold storage. The expression level of genes could further reflect the change rule of enzymes and provide information at a transcriptional level [33]. In this study, the expression levels of *ZlH^+^-ATPase*, *ZlNa^+^-K^+^-ATPase*, *ZlCa^2+^-ATPase*, *Zl**MDH*, *ZlSDH,* and *ZlCCO* of the water bamboo shoots during cold storage indicated that the control shoots exhibited the lowest expression level of all these genes and the NO + MAP-treated shoots showed the highest. The NO-treated shoots possessed higher expression levels of *ZlH^+^-ATPase*, *Zl**CCO* than the MAP-treated shoots. These results indicated that NO and MAP could increase the expression level of genes involved in the TCA cycle and mitochondrial electron transport, which could guarantee adequate energy supply to plant cells during cold storage, and the NO + MAP-treated shoots showed the most healthy and sufficient energy supply chain at a transcriptional level. The gene expression change pattern of *ZlH^+^-ATPase*, *ZlNa^+^-K^+^-ATPase*, *ZlCa^2+^-ATPase*, *Zl**MDH*, *ZlSDH,* and *ZlCCO* were similar with the change pattern of enzyme activity (Figure 5, Figure 6 and Figure 7), indicating that these enzymes were directly controlled by gene transcription, which further confirmed the results of enzyme activity. Taken together, these results indicate that the NO + MAP treatment improves the postharvest quality of water bamboo shoots by enhancing their mitochondrial energy metabolism. MAP could further improve the preservation effect of NO on water bamboo shoots and maintain its energy status because the change patterns of respiratory rate, ATP content, and energy charge in the NO + MAP water bamboo shoots were more like NO-treated material than MAP, this phenomenon implied that the NO treatment before MAP changed its physiological response to postharvest deterioration and this alteration lasted through MAP treatment and also indicated that NO was the main effective factor in the combination of NO + MAP.

The postharvest preservation of water bamboo is of great importance in its production and sale. Evidently, in this research, the NO + MAP treatment was an efficient preservation method for water bamboo, and was significantly better than a single treatment method. The combination of NO and MAP has great application prospects because it is efficient, simple, easy to use, and non-toxic. To further improve the preservation effect of NO + MAP, other treatments such as hot air and nano-material wrapping after NO treatment should be tried; this should also be a future research direction.

## 4. Materials and Methods

### 4.1. Plant Materials

Water bamboo shoots were collected from the plants that were grown in the experimental field of Yangzhou University, Yangzhou, Jiangsu, China, and transferred to the laboratory within 2 h.

The uniform shoots without mechanical damages or disease were selected. Finally, 960 shoots were randomly divided into four groups.

### 4.2. Treatments

The shoots of the water bamboo were separated into four groups for the control, NO, MAP, and combination of NO and MAP, respectively. The method of NO treatment was as described by Qi [3]. Briefly, the shoots were treated with NO (30 μL L^−1^) for 4 h at room temperature. For the MAP treatment, the water bamboo shoots were put into sealed low-density polyethylene (LDPE) bags (28 cm × 40 cm). For the NO + MAP treatment, the shoots were immediately sealed in LDPE bags after NO treatment. The treated and control shoots were then stored at 4 °C and 100% relative humidity for 28 d. The samples were collected before treatment and every 7 d during storage.

### 4.3. Weight Loss, Firmness, and Respiration Rate Assays

Percent weight loss was calculated at each sampling time. The firmness of water bamboo shoots was determined by a texture analyzer (TA-XT2i, Stable Micro Systems Ltd., Godalming, UK), according to the method of Qi [3].

Respiration rate was analyzed as described by Deng [34]. Thirty water bamboo shoots were randomly sampled and put into three sealed containers (each containing 10 shoots) that contained 20 mL NaOH solution (0.4 M) at room temperature for 1 h. The contents of the containers were mixed with 5 mL saturated BaCl_2_ solution and three drops of phenolphthalein. The solution was titrated with 0.02 M oxalic acid until the red color disappear. The respiration rate was expressed as g CO_2_ kg^−1^ h^−1^.

### 4.4. ATP and Energy Charge Measurement

ATP, ADP, and AMP contents were measured as described by Li with some modifications [6]. Briefly, 3 g water bamboo shoots was ground in liquid nitrogen, homogenized with 25 mL perchloric acid (0.6 M), and then centrifuged at 16,000× *g* for 12 min at 4 °C. The supernatant was clarified by passing it through an MCE syringe filter and its pH was adjusted to pH 6.5–6.8 by KOH (1 M). The solution was filtered again with an MCE syringe filter. A high-performance liquid chromatography system (LC-2100, Shimadzu Corporation, Kyoto, Japan) with a reverse-phase Luna 5 μm C18 column and ultraviolet detection was applied to measure ATP, ADP, and AMP contents. Energy charge was calculated by the formula (ATP +1/2ADP)/(ATP +ADP +AMP) [6].

### 4.5. Mitochondrial Ultrastructure 

The mitochondrial ultrastructure was observed as described by Li, with minor modifications [35]. Small pieces (1 mm × 1 mm × 1 mm) of water bamboo shoots were fixed in a mixture of 3% (*v*/*v*) glutaraldehyde for 24 h at room temperature. The samples were subsequently fixed with 1% osmic acid (*v*/*v*) for 5 h at room temperature, rinsed 3 times in 0.1 M phosphate buffer (pH 7.4), dehydrated in a series of ethanol, and finally embedded in spur resin. Ultrathin sections were cut, stained with 2% uranyl acetate and lead citrate for 15 min, respectively, and observed by using an HT7700 transmission electron microscope (Hitachi, Tokyo, Japan) at 80 kV.

### 4.6. The Activities of H^+^-ATPase, Ca^2+^-ATPase, and Na^+^-K^+^- ATPase

The activities of H^+^-ATPase, Ca^2+^-ATPase, and Na^+^-K^+^-ATPase were analyzed by using Ca^2+^-ATPase and Na^+^-K^+^–ATPase assay kit (Nanjing Jiancheng, Nanjing, China), following the manufacturer’s protocols.

### 4.7. Measurement of Ca^2+^ Concentration

Ca^2+^ content was assayed with a ZCA-1000 atomic absorption spectrometer (Zhonghe Cetong, Beijing, China).

### 4.8. Mitochondria Extraction and Enzyme Extraction

The mitochondria of water bamboo shoots were extracted according to the method of Kan, with minor modifications [36]. In brief, a frozen sample (10 g) was homogenized with 20 mL extraction buffer (0.05 M Tris, 0.25 M sucrose, 0.3 M mannitol, 1 mM EDTA, and 0.5% PVP, pH 7.5) on ice. The crude homogenate was filtered through gauze and 10 mL solution was collected. The filtrate was centrifuged for 10 min at 4000× *g* and the supernatant was collected and centrifuged for 20 min at 14,000× *g*; the mitochondria sediment was resuspended in washing buffer (0.01 M Tris, 0.3 M mannitol, 0.25 M sucrose, and 1 mM EDTA, pH 7.2). The solution was centrifuged for 10 min at 1000 × g and the supernatant were collected and centrifuged for 20 min at 14,000× *g*. The sediment was resuspended in 1.5 mL washing buffer. Protein contents were determined as described by Bradford and the final precipitate preserved for MDH, SDH, and CCO determinations [37].

Mitochondrial MDH activity was analyzed according to Lü, with some modifications [37]. Briefly, 0.1 mL mitochondrial extract, 0.1 mL NADH solution, and 2.7 mL phosphate buffer (pH 7.6) were mixed and the absorbance at 340 nm was immediately measured. Then, 0.1 mL acetyl oxalate was added to the reaction solution and the absorbance was determined 5 times at 30 s intervals. The activity of MDH was expressed as U g^−1^ protein, where one U is defined as an increase of 0.01 OD_340_ per min.

Succinic dehydrogenase (SDH) activity was assayed as described by Ackrell, with some modifications [38]. A solution with a final concentration of 0.2 M phosphate buffer (pH 7.4), 0.2 M sodium succinate (pH 7.4), 0.9 mM 2, 6-dichlorphenol indophenol, and crude mitochondrion extract was incubated for 5 min at 30 °C. Three hundred microliters of 0.33% 5-methylphenaziniummethosulfate was added to 1.7 mL reaction solution before determination. The absorbance was determined at 600 nm and the activity of SDH was expressed as U g-1 protein, where one U is defined as an increase of 0.01 OD_340_ per min.

CCO activity was analyzed as described by Errede [39]. The assay solution consisted of 2% TritonX-100, 0.2 M K-phosphate buffer, and 500 μL of the mitochondrial extract. The mixed solution was kept for 5 min at 30 °C. Cytochrome c was added to a final concentration of 20 g L^−1^ before determination. The absorbance was determined at 510 nm and the activity of CCO was expressed as U g^−1^ protein, where one U is defined as an increase of 0.01 OD_510_ per min.

### 4.9. RNA Extraction and Quantitative Real-Time PCR Analysis

The shoots of the water bamboo before cold storage and 0, 7, 14, 21, and 28 days after storage were collected. Primer design, RNA extraction, and quantitative real-time PCR (qPCR) were applied as previously described by Qi [33]. The relative gene expression used the 2^−ΔΔCt^ method for relative quantification. Three biological replications were considered for gene expression analysis for each cDNA. The primer sequences were shown in Appendix A.

### 4.10. Statistical Analyses

Each experiment was conducted with a randomized block design. The data were expressed as mean ± standard deviation (SD). ANOVA was analyzed based on SPSS 22.0 (SPSS Inc., Chicago, IL, USA). Tukey’s test was applied to determine the differences between means.

## 5. Conclusions

An understanding of the physiological changes that occur in stored water bamboo shoots is necessary to improve their postharvest quality. The present results indicate that softness and weight loss are the main problems associated with senescence during the postharvest storage of water bamboo shoots. MAP and NO efficiently inhibit the deterioration of stored water bamboo shoots by enhancing the mitochondrial energy metabolism. MAP improved the effect of NO in extending the postharvest life of water bamboo shoots.

## Figures and Tables

**Figure 1 ijms-23-01607-f001:**
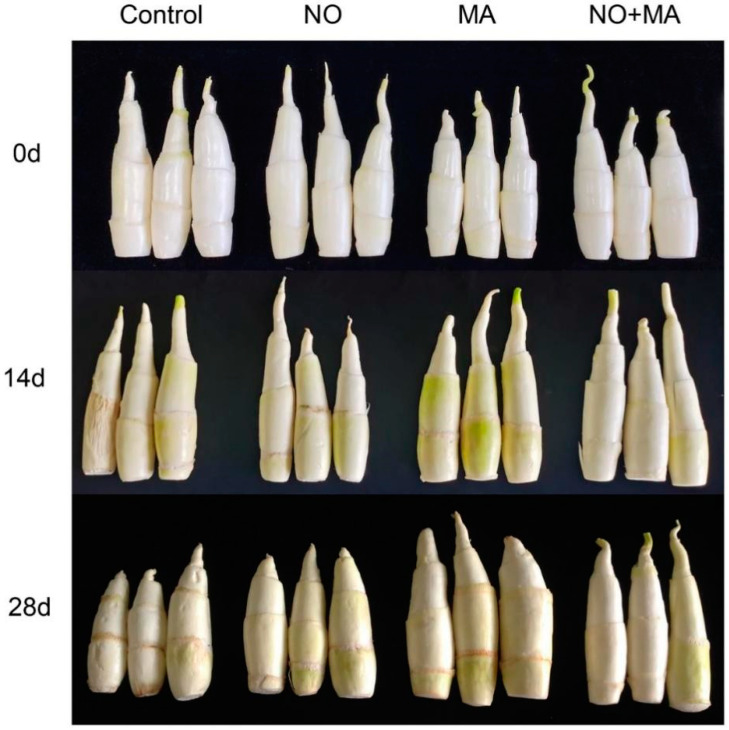
Effects of NO and MAP treatment on the color and skin-wilting of water bamboo shoot stored at 4 °C for 0, 14, and 28 d.

**Figure 2 ijms-23-01607-f002:**
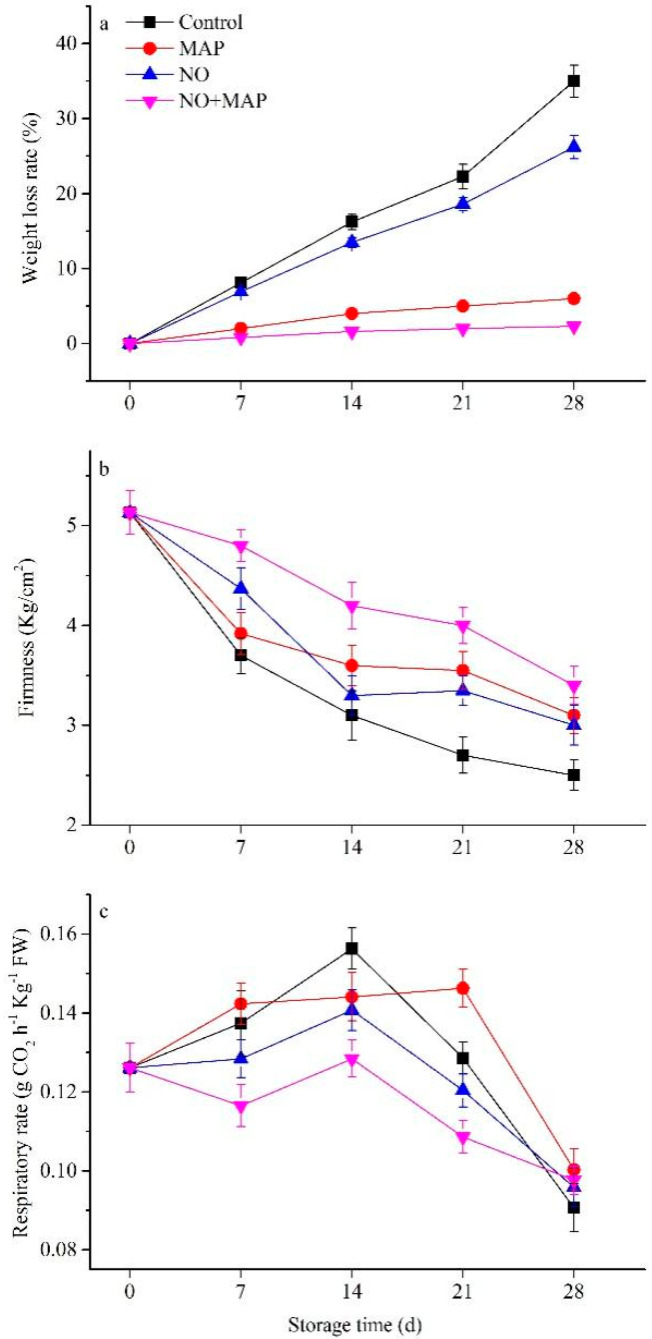
Effects of NO and MAP treatment on weight loss (**a**), firmness (**b**), and respiratory rate (**c**) in water bamboo shoots stored at 4 °C for 0, 7, 14, 21, and 28 d. Values are presented as mean ± SD.

**Figure 3 ijms-23-01607-f003:**
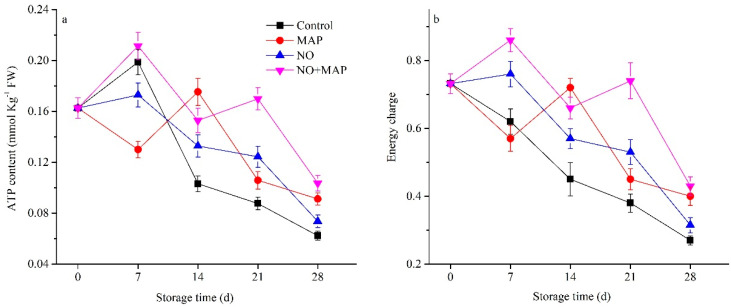
Effects of NO and MAP treatment on ATP (**a**) and energy charge (**b**) in water bamboo shoots stored at 4 °C for 0, 7, 14, 21, and 28 d. Values are presented as mean ± SD.

**Figure 4 ijms-23-01607-f004:**
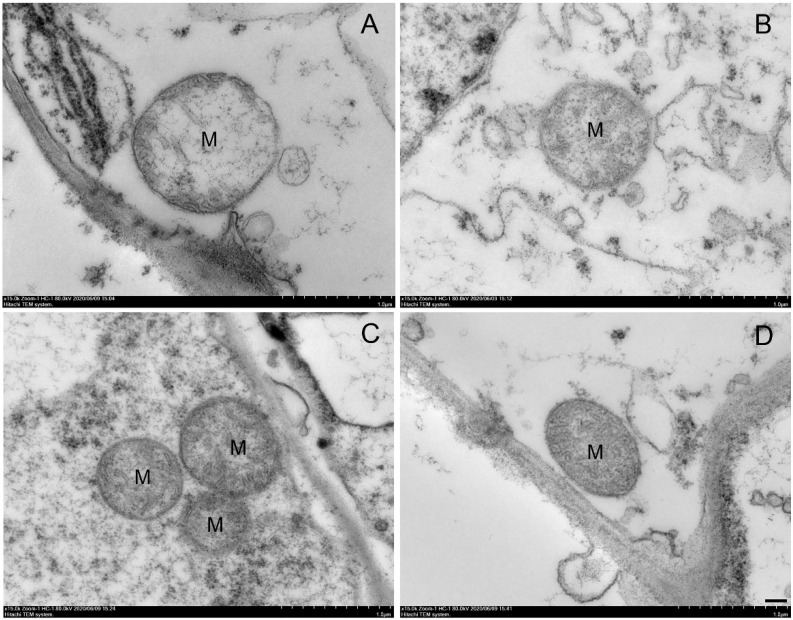
Effects of NO and MAP treatment on mitochondrial ultrastructure in control (**A**), NO (**B**), MAP (**C**), and NO + MAP (**D**) water bamboo shoots stored at 4 °C for 28 d. M represents mitochondria. Pictures show 15,000 × magnification. Bar represent 1 μm.

**Figure 5 ijms-23-01607-f005:**
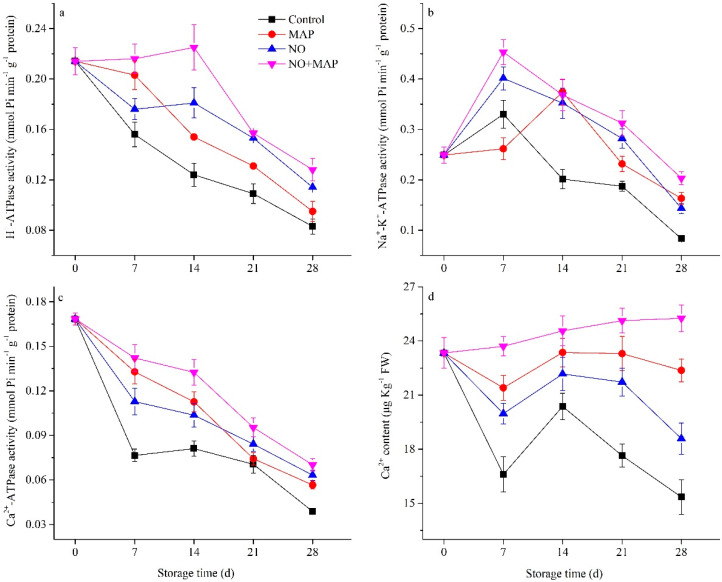
Effects of NO and MAP treatment on activities of H^+^-ATPase (**a**), Na^+^-K^+^-ATPase (**b**), Ca^2+^-ATPase (**c**), and Ca^2+^ content (**d**) in water bamboo shoots stored at 4 °C for 0, 7, 14, 21, and 28 d. Values are presented as mean ± SD.

**Figure 6 ijms-23-01607-f006:**
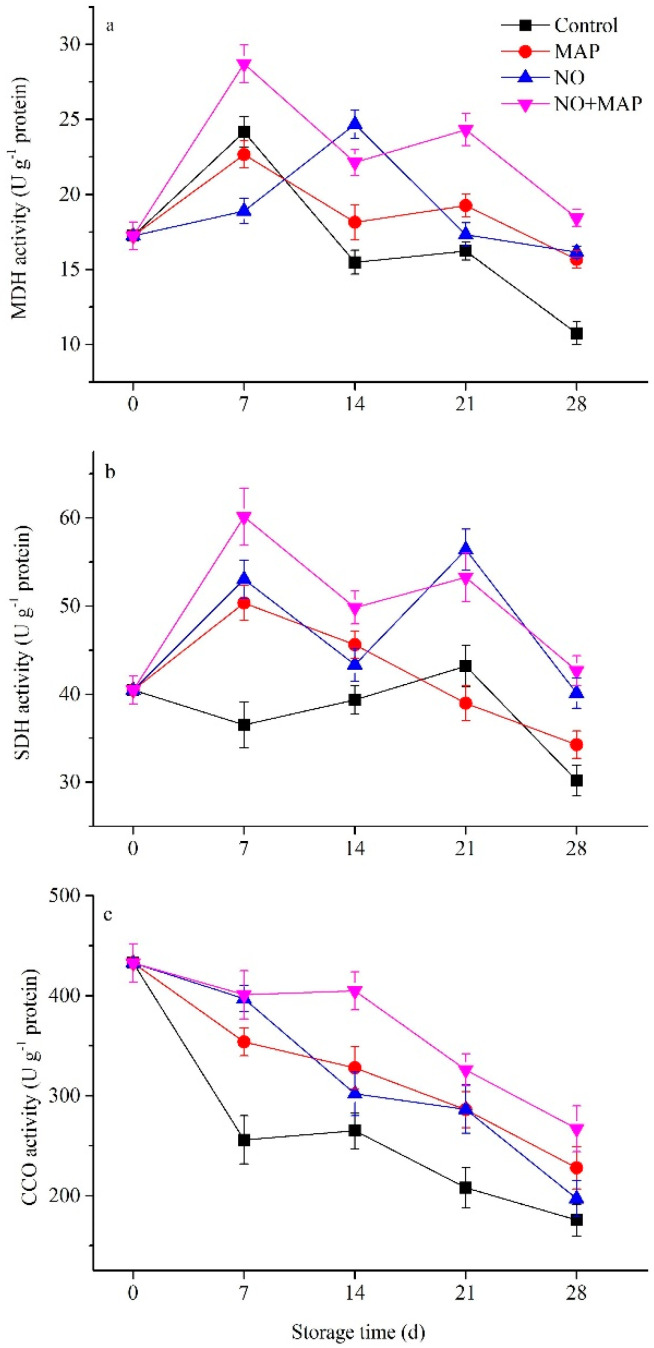
Effects of NO and MAP treatment on activities of MDH (**a**), SDH (**b**), and CCO (**c**) in water bamboo shoots stored at 4 °C for 0, 7, 14, 21, and 28 d. Values are presented as mean ± SD.

**Figure 7 ijms-23-01607-f007:**
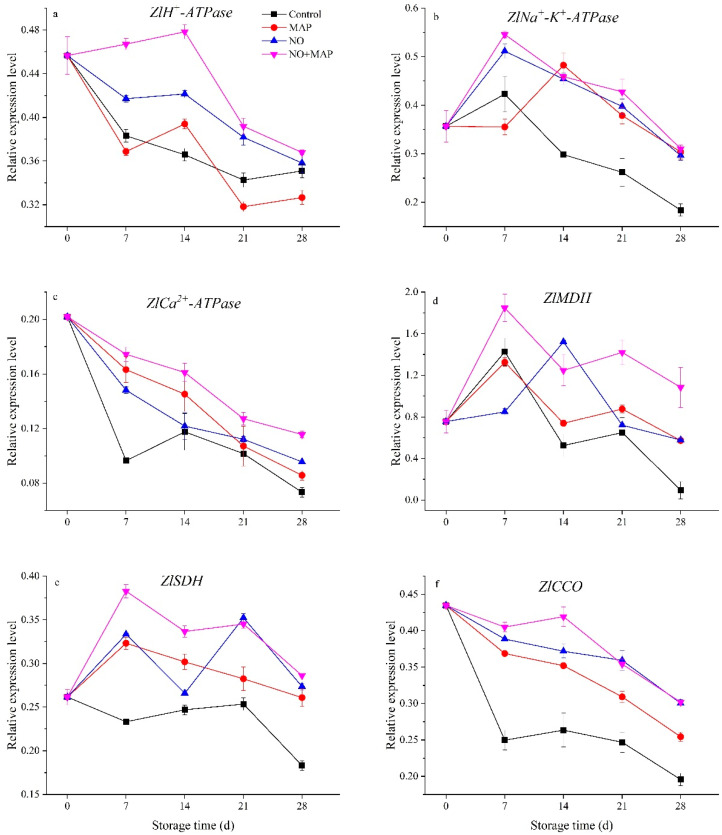
Effects of NO and MAP treatment on the expression of *ZlH^+^- ATPase* (**a**), *ZlNa^+^-K^+^- ATPase* (**b**), *ZlCa^2+^- ATPase* (**c**), *ZlMDH* (**d**), *ZlSDH* (**e**), and *ZlCCO* (**f**) in water bamboo shoots stored at 4 °C for 0, 7, 14, 21, and 28 d. Values are presented as mean ± SD.

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
