# Peer review of "Nitric Oxide Extends the Postharvest Life of Water Bamboo Shoots Partly by Maintaining Mitochondrial Structure and Energy Metabolism"

_ijms, 2022, doi:10.3390/ijms23031607_

Round 1

Reviewer 1 Report

Editor-in-Chief

Dear Sir

Manuscript title: Nitric oxide extends the postharvest life of water bamboo shoots partly by maintaining mitochondrial structure and energy metabolism

Manuscript Number: ijms-1523495

Dear Editor,

Please find evolution report of the manuscript entitled “Nitric oxide extends the postharvest life of water bamboo shoots partly by maintaining mitochondrial structure and energy metabolism”.

I went thoroughly on this manuscript, and my comments mentioned below.

The authors not well addressed the effect of NO and MAP on the quality of bamboo shoots during long-term storage. The experimental design and manuscript writing is just average. There is no novelty in this current form, also it’s not reached up to the standards of IJMS. Therefore, my recommendation is to reject the manuscript.

Author Response

Dear reviewer and editor,

Thank you for your kind comments, we thoroughly revised the paper, added some data, analyzed the detail of results and discussed the result more deeply, so we believe we increased the scientificity of the paper. More details please see the revised manuscript. Thank you.

Best wishes.

Qian

Reviewer 2 Report

The authors perform a study on interventions into postharvest stored crop - bamboo shoots  in order to prevent their natural decay in the storage unit. In this manuscript the authors try a combination of Nitrous Oxide and MAP and as controls, each of them alone. 

The manuscript need to undergo a thorough revision in English, as well as deep revision. It is very hard to understand what is the merit of the manuscript. The introduction lacks sufficient details on previous studies on Bamboo shoot. Just tells us what each of the methods -NO and MAP doesnt help. 

The results are described in short, without explaining what is going on in the study. Some captions are not related to the shown picture (Figure 4), and others are not explained what is seen in the figure (Figure 4 - what is the difference between the four TEM panels? ). The main picture with the bamboo shoot show some benefit for the product, but really its not that good, and at the 28d there is virtually no difference. 

Discussion is lacking with strict conclusions. I was not impressed by the Data, there is a difference in enzymatic activity between each treatmetn and control, but there is hardly any difference between the treatments, so its not understood what is the advantage of this or another method.

Finally, statistics is lacking. There is no repeated measures ANOVA for differences between points in time in each treatment, also it is not understood how many repeats were taken to ANOVA, and if ANOVA is at all parametric or non-parametric and how it was checked.

In Summary, the manuscript is written in a very specialized way which an outsider to this scientific field will not understand. I suggest the authors to think again about the target of their research and how to convey it better for the reader.

Author Response

Dear reviewer and editor,

Thank you for your kind comments, we thoroughly revised the paper, added some data, described the detail of results and discussed the result more deeply, and also revise the language.

so we believe we increased the scientificity and legibilityof the paper. More details please see the revised manuscript. Thank you.

Best wishes.

Qian

Round 2

Reviewer 1 Report

Dear Authors, 

 I am rejecting the manuscript in its current form.

Thank you

Author Response

Dear reviewer,

Thank you for your comment on our paper, although your suggestion is rejection. We research on the preservation effect of nitric oxide on water bamboo for a few years, and the energy metabolism is a very important part of the preservation mechanism of nitric oxide. The combination of nitric oxide and modified atmosphere packaging could significantly improve the quality of water bamboo under single treatment. This combination is an easy and practical method for the preservation operation of water bamboo, and also suggested for agricultural practice and application. The effect of nitric oxide in this combination, also analyzed through the energy metabolism of water bamboo during cold storage. The result is informative, the result analysis is correct and detailed, the discussion is also comprehensive and in depth, so we believe our research is worth publishing and reading. We kindly request you give us another chance.

Best regards

                                                                                    Qian

                                                                         Yangzhou University

Reviewer 2 Report

The authors improved substantially the presentation of the study and the description of the novelty of their results. They discuss nicely how their study relate to other research in this scientific area. 

I am not sure that this journal is the best fit for this manuscript, and think that it should be transferred to a more appropriate journal for example - Post Harvest Biology and Technology (higher impact factor :) ). 

Nevertheless, it is now much better in conveying the message.

Good job. 

Minor revision: add a paragraph at the end what are the consequences in large, and a summary on the benefits of the methods in general and a look into the future. 

Conclusion should be changed in location after discussion, and not after methods. Also, the manuscript could benefit from another pass with professional English editor.

Good luck !

Author Response

Dear reviewer,

Thank you for your comment, we minor revised our paper followed your suggestion, details are as follow.

The authors improved substantially the presentation of the study and the description of the novelty of their results. They discuss nicely how their study relate to other research in this scientific area.

I am not sure that this journal is the best fit for this manuscript, and think that it should be transferred to a more appropriate journal for example - Post Harvest Biology and Technology (higher impact factor :) ).

Nevertheless, it is now much better in conveying the message.

Good job.

Minor revision: add a paragraph at the end what are the consequences in large, and a summary on the benefits of the methods in general and a look into the future.

Answer: we already added a paragraph and stated the benefits of this method.

Conclusion should be changed in location after discussion, and not after methods. Also, the manuscript could benefit from another pass with professional English editor.

Answer: We already changed the location of conclusion, and we also revised the language.

Good luck !

Thank you for your suggestion, we already improved our paper, please find the details in the minor revised paper.

Best regards

Qian

Yangzhou University

Round 3

Reviewer 1 Report

Dear Authors,

I went thoroughly on this current manuscript style and results. I recommend your manuscript for acceptance for publication in its current form.

Thank you